# The magnitude of undernutrition and associated factors among adult chronic kidney disease patients in selected hospitals of Addis Ababa, Ethiopia

**Mahder Asefa[1], Amene Abebe[2], Behailu Balcha[3], Daniel Baza[ID][4]***

**1** Department of Human Nutrition, College of Medicine and Health Science, Wolaita Sodo University, Wolaita Sodo, Ethiopia, **2** Department of Reproductive Health and Human Nutrition, College of Medicine and Health Science, Wolaita Sodo University, Wolaita Sodo, Ethiopia, **3** Department of Public Health, College of Medicine and Health Science, Wolaita Sodo University, Wolaita Sodo, Ethiopia, **4** Department of Pediatrics and Neonatal Nursing, College of Medicine and Health Science, Wolaita Sodo University, Wolaita Sodo, Ethiopia

\* danielbaza9@gmail.com

**Data Availability Statement:** All relevant data are within the paper and its Supporting Information files.

**Funding:** This study did not receive any fund.

## Abstract

### Background

Undernutrition is a common comorbidity in chronic kidney disease patients which augments the progression of the disease to an end-stage renal disease, renal dysfunction and related morbidity and mortality. However, in Ethiopia, there is a dearth of research evidence in this regard. Therefore, this study aimed to assess the magnitude of undernutrition and its associated factors among adult chronic kidney disease patients.

### Methods

An institution-based cross-sectional study was conducted in selected hospitals of Addis Ababa from May to August 2018. Data were collected by structured and pretested questionnaires. Patients' charts were reviewed from their medical profiles. Body mass index was calculated from anthropometric measurements using calibrated instruments. Serum albumin level was determined by reference laboratory standard procedure. Data were entered into Epi- data version 3.1 and exported to SPSS version 21 for analysis. Descriptive statistics were calculated and presented by tables, graphs and texts. Binary and multivariable logistic regression analyses were computed and the level of statistical significance was declared at p-value <0.05.

### Results

From the total sample size of 403 participants, 371 were involved in the study. The prevalence of undernutrition (BMI<18.5) among adult chronic kidney disease patients was 43.1% (95% CI: 38%-48%). Undernutrition (BMI<18.5) was significantly higher among patients with diabetic nephropathy [AOR = 2.00, 95% CI, 1.09–2.66], serum albumin value less than

**Competing interests:** The authors declared that they have no conflict of interest.

**Abbreviations:** AOR, Adjusted odds ratio; CI, confidence interval; IV, four; V, five; CKD, chronic kidney disease; g/dl, gram per deciliter; ESRD, end stage renal disease; DM, diabetes mellitus; BMI, body mass index; GFR, glomerular filtration rate; OPD, out -patient department; KG, kilogram; cm, centimeter; HIV, human immuno virus.

3.8g/dl [AOR = 4.21: CI, 2.07–5.07], recently diagnosed with diabetes mellitus [AOR = 2.36, 95% CI, 1.03–3.14] and stage V chronic kidney disease [AOR = 3.25:95% CI, 1.00–3.87].

## Conclusion

Undernutrition in chronic kidney disease patients was significantly higher among patients with diabetic nephropathy, patients on stage V chronic kidney disease, recently diagnosed with diabetes mellitus and serum albumin value less than 3.8g/dl.

## Background

Chronic kidney disease is defined as abnormalities of kidney structure or function present for more than three consecutive months [1]. Although the incidence and prevalence of chronic kidney disease (CKD) have upsurge globally, the trend is not similar between developed and developing countries. Evidence from sub-Saharan Africa suggests 12–23% of adults have Chronic Kidney Disease (CKD) and are therefore at risk of developing (End Stage Renal Diseases) ESRD [2–4].

Undernutrition among CKD patients is found to be 29% using body mass index (BMI) as a diagnostic tool. It is a major comorbidity in CKD and ESRD patients as equal to others like hypertension, Diabetes Mellitus (DM) and cardiovascular diseases [5]. Undernutrition in CKD patients is one of the early complications which indicate the poor prognosis of the disease. Moreover, it speed up prognosis of the disease to ESRD which does not have options of treatment other than lifelong dialysis or renal replacement therapy [6].

Undernutrition is common in CKD patients but it is an enormously dominant comorbidity in ESRD patients. ESRD can be attributable to multiple factors including lower intakes of protein and calorie, diabetes mellitus, hypertension, lack of exercise, age, losses of nutrients, family history of the same disease, frequent hospitalization, gastrointestinal diseases, frequent dialysis, stage of GFR and multiple medications [7–9].

Limited evidence suggests undernutrition is a rapidly growing predisposing factor for several underlying diseases which complicates the care and treatment of CKD patients. However, nutritional status of CKD patients is left unknown fully [7–10]. Therefore, this study aimed to assess the undernutrition and associated factors among adult CKD patients in selected hospitals of Addis Ababa, Ethiopia.

## Methods and materials

### Study design, period and setting

An institution-based cross-sectional study was conducted from May to August 2018. The study was carried out at Saint Paul's and Zewditu Memorial Hospitals in Addis Ababa, Ethiopia. The two hospitals have been selected because of their well-established renal outpatient departments and mostly referred by other health institutions from all over the country. Saint Paul's hospital is in Addis Ketema Kifle Ketema and was reformed as a medical college in 2007. It has more than 13 departments; of which internal medicine is one where renal care is practiced and has a capacity of around 1440 patients in a year. It has three outpatient departments (OPD) working twice per week. Zewditu Memorial Hospital is one of the most popular hospitals in Addis Ababa which is in Arada-Kifleketema, Addis Ababa, Ethiopia. Currently, the hospital provides many services including dialysis. The hospital has one renal OPD working twice per week.

## Population

All adult chronic kidney disease patients who are on follow-up care in the selected two hospitals during the study period were the source population of the study. CKD patients on follow-up care for at least three months were included in the study. Patients diagnosed with liver disease and who are on dialysis were excluded from the study.

## Sample size determination and sampling procedure

By using single population proportion formula and considering confidence level /Z/ of 95%, marginal of error 5%, a reasonable estimate for the proportion of undernutrition in CKD patients (P = 0.5) and adding a none response rate of 5%, a total sample size of 403 CKD patients was obtained. The total sample size was allocated using average number of CKD patients visiting renal OPDs in the two selected hospitals. The average number of patients visiting OPD at Saint Paul's hospital in one month was 100 and Zewditu Memorial Hospital was 50. The total numbers of patients being treated in the two hospitals in one month was estimated to be 150. Accordingly, from Saint Paul's hospital = 403*100/150 = 268 patients and from Zewditu Memorial Hospital = 403*50/150 = 135 patients were included in the study. Every 3rd and 5th patient has been included in the study using a systematic sampling technique after determining the sampling interval of 403/268 = 3 in Saint Paul's and 403/135 = 5 in Zewditu Memorial Hospital. The first participant was recruited by using the lottery method in each hospital.

# Variables of the study

## Dependent variable

Undernutrition in CKD patients (Yes/No).

## Independent variables

**Socio-demographic factors.** Age, sex, place of residence, occupation, marital status, income, educational level, dietary intake.

**Clinical factors.** Stage of CKD, duration of illness, cause of disease, comorbidity, family history of the disease, type of disease, serum albumin level.

**Lifestyle factors.** Exercise, smoking, alcoholism, sleep patterns.

**Nutritional factors.** Frequency of meal, dietary diversity, nutrition counseling, place of meal preparation, patient appetite.

## Operational definitions

**Undernutrition.** Chronic kidney disease patients whose measured BMI level of less than 18.5 [11].

**Chronic kidney disease.** Confirmed chronic abnormalities of kidney structure or function, present for >3 months with implications for health [12].

**Glomerulonephritis.** CKD patients with persistent proteinuria and occasional hematuria [13].

**Stages of CKD.** Stage 1CKD (GFR $\geq$ 90 mls/min), stage 2 CKD (GFR 60-89mls/min), stage 3 CKD (GFR = 59–30 mls/min), stage 4 CKD (GFR = 15–29 mls/ min) and stage 5 CKD (GFR < 15mls/min) [14].

**Hypoalbuminemia.** Serum albumin level less than 3.8 g/dl.

## Data collection procedures

Socio-demographic and dietary data were collected by professional nurses and phlebotomists using interviewer-administered, pre-tested and structured questionnaires. Clinical data on the cause of the disease, stage of GFR and comorbidities were retrieved from patients' medical charts.

Measurement of weight was recorded using a standard digital scale. The digital scale read-out was checked reading of zero before measuring weight. Measurements were done after participants standing on the center of the weight scale platform dressing light clothes. Weight measurement results were recorded to the nearest of 0.1 kg.

Height was measured using a standard stadiometer. The study participants were asked to remove their shoes and stand erect. The heels of the feet were placed together with both heels touching the base of the vertical board. The head is maintained in the Frankfort Horizontal Plane position while the examiner lowers the horizontal bar snugly to the crown of the head with sufficient pressure to compress the hair. Results were recorded to the nearest 0.1cm. Participants' BMI was calculated using their weight and height measurements; weight/ (height) $^2$(Kg/m$^2$).

Serum albumin was determined by the Ethiopian Public Health Institute national HIV reference laboratory standard procedure. Appropriate precautions were undertaken to ensure the safe collection of blood samples. Ten milliliters of venous blood was collected following standard aseptic techniques and then centrifuged for five minutes at 3000 revolutions per minute. The separated serum was restored in a separate tube (Nunk tube) and was stored in a deep freezer and the serum albumin level was determined.

Dietary intake data were gathered using 24 hours dietary recall technique. Patients were asked to list out all the foods and drinks they had in the last 24 hours and the information has been recorded.

## Data collection tools and data quality assurance

The data were collected by using pre-tested questionnaire. The questionnaire was pre-tested at Minilik Hospital on 5% of CKD patients before actual data collection. The questionnaire was translated from English to the Amharic language by a person who is fluent in both languages. Three days of training was given for data collectors. Calibration of weight and height measuring instruments was done after each measurement then the data were entered in ENA SMART software to see relative Technical Error of Measurements (TEM). The TEM output was compared with the acceptable range for relative TEM using beginner anthropometric levels for inter-examiners and was found to be in the acceptable range, < 2.0%.

Multicollinearity test was done and reported by Variance Inflation Factor (VIF) and found in the acceptable range <5. Specimen collection and preparation were done by experienced clinical laboratory professionals using standard equipment and following standard procedures. The overall clinical data collection was supervised by a Nephrologist.

## Data analysis and management

The collected data were entered into Epi- data Version 3.1 and exported to SPSS version 21 for analysis. Frequencies, proportion, summary statistics and cross-tabulation of study variables was done and presented in simple frequency tables, graphs and texts. The assumptions for binary logistic regressions were first checked and then bivariable analysis was carried out to identify candidate variables at p-value <0.25 for multivariable analysis. Multivariable logistic regression analysis was done using those candidate variables to identify the statistically significant and independent associations between dependent and independent variables. The

strength of associations was presented using an adjusted odds ratio with its 95% CI. Finally, variables whose p-value < 0.05 in multivariable logistic regression model were declared as statistically significantly associated.

## Ethics approval and consent to participate

The study was approved by the Research Ethics Review Committee, College of Health Science and Medicine, Wolaita Sodo University. Participants were informed clearly about the purpose, risks and benefits of the study. Informed written consent was obtained from each participant. Confidentiality of participants' information was kept throughout the research process. Personal privacy and cultural norms were respected. All biomedical waste products were discarded maintaining standard safety protocols.

## Results

### Socio demographic characters

From the total sample size of 403 study participants, 371 were actually involved in the study making the response rate of 92%. Out of the total study partakers, 231 (62.3%) were males the rest were females. The mean age of the participants was 43 years with a standard deviation of ±14 years. A high proportion 276 (74.4%) of the participants were urban dwellers and 110 (29.7%) of them were a government employees. Twenty-four (6.5%) of the respondents reported they had lost their jobs because of the disease. Regarding educational status, the majority 307 (82.8%) attended above primary education and more than half 237 (63.9%) were married "Table 1".

**Table 1. Socio demographic characteristics of chronic kidney disease patients in Saint Paulo's and Zewditu Memorial Hospitals in Addis Ababa, Ethiopia, 2018.**

| Variable | Frequency | Percent (%) |
|---|---|---|
| **Sex (n = 371)** | | |
| Male | 231 | 62.3 |
| Female | 140 | 37.7 |
| **Age in years (n = 371)** | | |
| 18–24 | 32 | 8.6 |
| 25–54 | 257 | 69.3 |
| 55–64 | 52 | 14.0 |
| ≥65 | 30 | 8.1 |
| **Place of Residence (n = 371)** | | |
| Urban | 276 | 74.4 |
| Rural | 95 | 25.6 |
| **Occupation (n = 371)** | | |
| Farmer | 28 | 7.5 |
| Merchant | 40 | 10.8 |
| Daily laborer | 28 | 7.5 |
| Government employee | 110 | 29.7 |
| Non-government employee | 76 | 20.5 |
| Retired | 39 | 10.5 |
| Lost job | 24 | 6.5 |
| Others * | 26 | 7.0 |
| **Marital status(n = 371)** | | |

(*Continued*)

**Table 1.** (Continued)

| Variable | Frequency | Percent (%) |
|---|---|---|
| Married | 237 | 63.9 |
| Single | 103 | 27.8 |
| Widowed | 8 | 2.2 |
| Divorce | 23 | 6.2 |
| **Educational level (n = 371)** | | |
| No formal education | 64 | 17.3 |
| Primary | 129 | 34.8 |
| Secondary | 132 | 35.6 |
| Higher education | 46 | 12.4 |
| **Monthly income in ETB (n = 371)** | | |
| <700 | 94 | 25.3 |
| 701–2000 | 104 | 28 |
| 2001–5000 | 136 | 36.7 |
| >5001 | 37 | 10 |

* = Self-employed, house wives, religious workers

ETB = Ethiopian Birr

## Life style patterns of CKD patients in Saint Paulo's and Zewditu Memorial Hospitals

In this study, the majority 286 (77.1%) never drank alcohol, while 27 (7.3%) of them quit smoking. A high proportion of 341 (92%) of the participants did not have regular physical activity "Table 2".

## Prevalence of undernutrition and nutritional habits of chronic kidney disease patients in Saint Paul's and Zewditu Memorial Hospitals

The overall prevalence of undernutrition among the study participants was found to be 160 (43.1%) having a measured BMI level of less than 18.5. The mean BMI value was 22.8 with the

**Table 2. Life style patterns of CKD patients in Saint Paulo's and Zewditu Memorial Hospitals in Addis Ababa, Ethiopia, 2018.**

| Variable | Frequency | Percentage |
|---|---|---|
| **Drinking alcohol(n = 371)** | | |
| Never drink | 286 | 77.1 |
| Quit drinking | 84 | 22.6 |
| I drink now | 1 | .3 |
| **Smoking (n = 371)** | | |
| Never smoked | 343 | 92.5 |
| Quit Smoking | 27 | 7.3 |
| Currently smoking | 1 | .3 |
| **Physical activities(n = 371)** | | |
| Yes | 30 | 8.1 |
| No | 341 | 91.9 |
| **Sleep pattern(n = 371)** | | |
| Difficulty in falling asleep | 80 | 21.6 |
| Do not get enough sleep | 72 | 19.4 |
| No change | 219 | 59.0 |

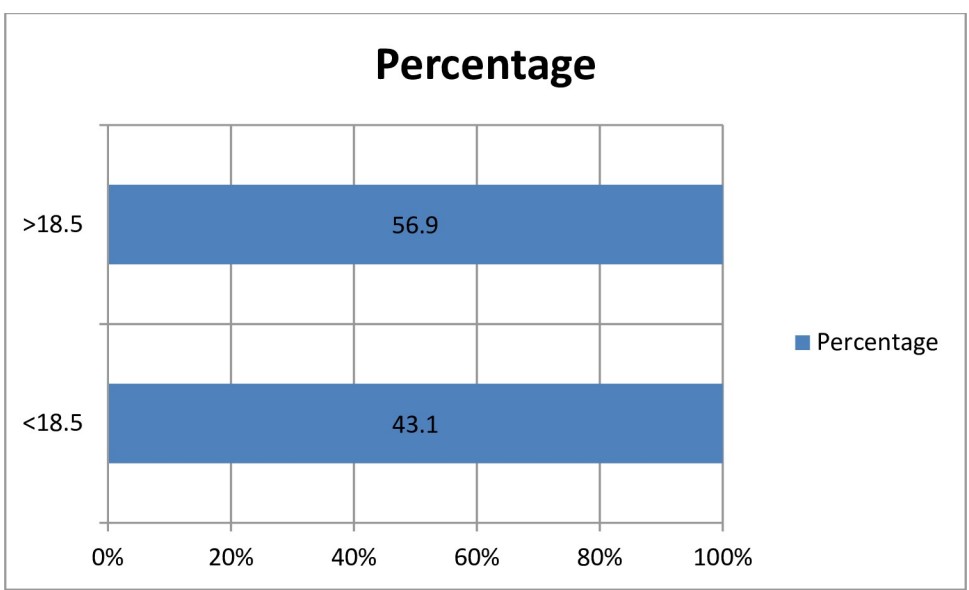

**Fig 1. BMI measures of chronic kidney disease patients in Saint Paulo's and Zewditu Memorial Hospitals in Addis Ababa, Ethiopia, 2018.**

standard deviation of ±3.5 "Fig 1". From the total study respondents, only 235 (63.3%) afforded eating meals three times a day. One hundred fourteen (30%) of the participants had provided nutritional counseling and 299 (80.6%) reported they prepare meals at their homes. Two hundred seventy one (73%) and 251 (67.7%) respondents had decreased appetite and weight respectively in the past 3 months before the study. Regarding food diversity, the majority 300 (80.9%) of the participants reported they eat less than or four food groups in the last 24 hours "Table 3".

## Clinical conditions of the study participants in Saint Paulo's and Zewditu Memorial Hospital, Addis Ababa, Ethiopia

From the total of 371 participants involved in the study, 195(52.5%) of them were diagnosed with CKD for less than 4 years. Diabetic nephropathy was found to be the predominant cause of CKD accounting for 166 (44.7%) and about 147 (39.6%) diagnosed with DM in the last three months before the study. Hypoalbuminemia (serum albumin level less than 3.8g/dl) was seen in 161 (43.4%) of the study participants "Table 4".

## The stages of CKD among study participants in Saint Paulo's and Zewditu Memorial Hospitals

Stages of chronic kidney disease were determined by glomerular filtration rate. The stage IV CKD was found in 138 (37%) and stage III in 133 (36%). The magnitude of stage I and stage II CKD was 59 (16%). Stage V was found in 41 (11%) of study participants "Fig 2".

## Factors associated with undernutrition among chronic kidney disease patients of Saint Paulo's and Zewditu Memorial Hospitals

In the binary logistic regression analysis: stage of the disease, cause of the disease, duration of the disease, meal pattern, serum albumin values and recent diagnosis with DM were significantly associated at the p-value of <0.25 and were candidate variables for multivariable logistic

**Table 3. Nutritional habits of chronic kidney disease patients in Saint Paulo's and Zewditu Memorial Hospitals in Addis Ababa, Ethiopia, 2018.**

| Variables | Frequency | Percentage |
|---|---|---|
| **Frequency of meal in a day (n = 371)** | | |
| 2meals a day | 66 | 17.8 |
| 3 meals a day | 235 | 63.3 |
| 4 meals a day | 70 | 18.9 |
| **Dietary diversity (n = 371)** | | |
| ≤4 | 300 | 80.9 |
| >4 | 71 | 19.1 |
| **Nutrition counseling (n = 371)** | | |
| Yes | 114 | 30.7 |
| **No** | 257 | 69.3 |
| **Where your meals prepared (n = 371)** | | |
| Home | 299 | 80.6 |
| Out of home | 72 | 19.4 |
| **Appetite (n = 371)** | | |
| Decreased | 271 | 73.0 |
| Increased | 13 | 3.5 |
| No change | 85 | 22.9 |
| Unnoticed | 2 | 0.5 |
| **Weight (n = 371)** | | |
| Decreased | 251 | 67.7 |
| Increased | 13 | 3.5 |
| No change | 107 | 28.9 |

regression analysis. In the multivariable logistic regression analysis: cause of the disease, stage of the disease, recent diagnosis with DM status and serum albumin value remain independently and significantly associated with undernutrition among CKD patients. Undernutrition was significantly higher in patients with diabetic nephropathy and hypertensive than those with glomerulonephritis [AOR = 2.00: 95% CI, 1.09–2.66] and [AOR = 2.13: 95% CI, 1.01–3.87] respectively. Stage V CKD patients were three times more likely to develop undernutrition than stage I and II patients [AOR = 3.25: 95% CI, 1.00–3.87]. Those patients in stage III [AOR = 2.01: 95% CI, 98–4.76] and stage IV [AOR = 2.03: 95% CI, 1.85–4.01] had more than two times higher chance of developing undernutrition than those in stage I & II.

Undernutrition was significantly higher among CKD patients whose serum albumin value less than 3.8 g/dl than those patients whose serum albumin value of greater than 3.8g/dl [AOR = 4.21: CI, 2.07–5.07] "Table 5".

## Discussion

This particular study has described the prevalence of undernutrition and associated factors among CKD Patients in Saint Paul's and Zewuditu Memorial Hospitals in Addis Ababa, Ethiopia. The prevalence of undernutrition among chronic kidney disease patients based on BMI level was found to be 160 (43.1%) out of the total sample size of 371 (100%). This finding is consistent with the studies done in teaching hospitals of Southern Nigeria, Nigeria and India, where the studies reported that the prevalence of undernutrition was 43.2%, 46.7%, and 42.7% [15–17]. The prevalence of undernutrition in the present study is not consistent with the study done in Jordan where the prevalence was 65% among chronic kidney disease patients [18].

**Table 4. Clinical conditions of chronic kidney disease patients in Saint Paulo's and Zewditu Memorial Hospitals, Addis Ababa, Ethiopia, 2018.**

| Variable | Frequency | Percentage |
|---|---|---|
| **Duration of disease (n = 371)** | | |
| 1–3 years | 195 | 52.5 |
| ≥4 Years | 176 | 47.5 |
| **Diagnosed with DM (n = 371)** | | |
| Yes | 147 | 39.6 |
| No | 224 | 60.4 |
| **Cause of Disease (n = 371)** | | |
| Diabetic nephropathy | 166 | 44.7 |
| Hypertensive nephropathy | 92 | 24.8 |
| Glomerulonephritis | 113 | 30.5 |
| **Family History (n = 371)** | | |
| Yes | 58 | 15.6 |
| No | 313 | 84.4 |
| **Diagnosed with HTN (n = 371)** | | |
| Yes | 190 | 51.2 |
| No | 181 | 48.8 |
| **Albumin level (n = 371)** | | |
| <3.8 g/dl | 161 | 43.4 |
| ≥3.8 g/dl | 210 | 56.6 |

However, it is much higher when compared to the study result of Bordeaux University Hospital, Bordeaux, France 24.2% [19]. The possible reasons for variation in the prevalence of under-nutrition can be attributed to differences in socio-demographic, economic, case mix, comorbidity and the differences in the diagnostic criteria used.

In the current study, diabetic nephropathy accounts for 166 (44.7%) of all CKD. This finding is inconsistent with the study result in Sub-Saharan Africa region where it was estimated to

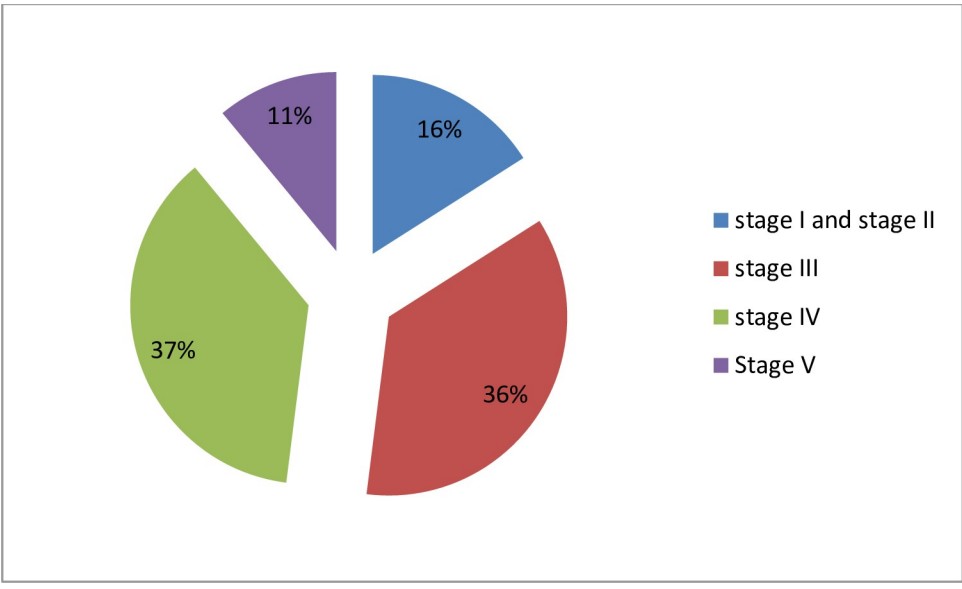

**Fig 2. Stages of CKD among study participants in Saint Paulo's and Zewditu Memorial Hospitals, 2018.**

**Table 5. Factors associated with undernutrition among CKD patients in Saint Paulo's and Zewditu Memorial Hospitals in Addis Ababa, Ethiopia, 2018.**

| Variables | BMI | | COR (95%CI) | AOR (95%CI) |
|---|---|---|---|---|
| | Greater than 18.5 (%) | Less than 18.5 (%) | | |
| **Cause of CKD (n = 371)** | | | | |
| Diabetic nephropathy | 98(59%) | 68(41%) | 2.19(1.28–3.56) | 2.00(1.09–2.66)* |
| Hypertension | 61(66.3%) | 31(33.7%) | 1.62(1.33–3.63) | 2.13(1.01–3.87)* |
| Glomerulonephritis | 52(46%) | 61(54%) | 1 | 1 |
| **Stage of CKD (n = 371)** | | | | |
| Stage 3 | 36(64.2%) | 20(35.8%) | 2.12(0.87–5.14) | 2.01(0.98–4.76) |
| Stage 4 | 80(59.7%) | 54(40.3%) | 2.95(1.34–6.53) | 2.03(1.85–4.01)* |
| Stage 5 | 23(56%) | 18(44%) | 2.18(1.23–4.82) | 3.25(1.00–3.87)* |
| Stage 1 and 2 | 72(51.4%) | 68(48.6%) | 1 | 1 |
| **Serum albumin value (n = 371)** | | | | |
| <3.8 g/dl | 72(44.7%) | 89(55.3%) | 2.13(1.78–3.25) | 4.21(2.07–5.07)* |
| ≥3.8 g/dl | 139(66.5%) | 71(33.5%) | 1 | 1 |
| **Recently diagnosed with DM (n = 371)** | | | | |
| Yes | 104(70.7%) | 43(29.3%) | 1.52(1.06–3.47) | 2.36(1.03–3.14)* |
| No | 107(47.7%) | 117(52.3%) | 1 | 1 |

* = variables significantly associated at p-value <0.05

be the cause of CKD in 6–16% [3, 16]. In the present study, of those patients who have been diagnosed with DM 166(45%), 68(41%) of them were undernourished (BMI <18.5). This finding is much higher than the study result of Bordeaux University Hospital, Bordeaux, France, where the study shows the minimum BMI among all the CKD study participants greater than 22.5 [19].

In the recent study, an increasing trend of undernutrition had been reported from stage I CKD to stage V and the highest prevalence 18 (44%) was seen in of stage V CKD patients. This finding is lower than the study result of Nigeria where the magnitude of 69% was reported in stage V CKD patients [16]. The disparity might be explained in terms of differences in socio-demographic and economic, socio-cultural and nutritional habits.

The risk of undernutrition was significantly higher in hypertensive and diabetic patients than those with glomerulonephritis [AOR = 2.00: 95% CI, 1.09–2.66] and [AOR = 2.13: 95% CI, 1.01–3.87] respectively. This result is not corroborated with the study result of the teaching hospital of Southern, Nigeria where the study reported no significant association between being hypertensive or diabetic and undernutrition in CKD patients [20]. The possible explanation for the difference might be due to the differences in sample size, study design, food taboos and cross-cultural variability in food selection.

The current study identified that being in stage V CKD was a risk for undernutrition than being in stage I and II [AOR = 3.25: 95% CI, 1.00–3.87] and the risk of undernutrition increases as the patients CKD stage progresses from stage I to stage V. This result is similar with the study results of Southern Nigeria and the Catholic University of Korea, South Korea [20–22].

In this study, patients recently diagnosed with DM were more than 2 times more likely to be undernourished than those who know their DM status three months before the study [AOR = 2.36: 95% CI, 1.03–3.14]. The possible explanation for this might be food restriction, being on anti- DM medications, stress due to recent diagnosis and complicated DM.

In this study, serum albumin value was also identified as a significant predictor of undernutrition among CKD patients. Undernutrition was significantly higher among CKD patients whose serum albumin value is less than 3.8g/dl when compared with their counterparts [AOR = 4.21: CI, 2.07–5.07]. This result is parallel to the study reports of teaching hospital of Southern Nigeria, USA, Stockholm, Sweden [21, 23, 24].

## Study strengths

This study has covered the prevalence of undernutrition among chronic kidney disease patients which was not well addressed previously. Moreover, in this study, the extent of anemia among CKD patients which is an important nutritional status indicator was determined by using the serum value of albumin. The generalizability of the study findings was high because the hospitals included in the study serve much population which comes from throughout Addis Ababa city and its surroundings.

## Study limitations

This study has limitations. First, as the study sample consisted of adult CKD patients who come to selected health facilities and therefore we cannot generalize our findings to other districts elsewhere in Ethiopia or other sub-Saharan developing countries. Second, this study has assessed undernutrition using only BMI measures which may miss the comprehensive nutritional status. Information was collected on exposures and outcomes simultaneously, thus causal relationships are difficult to establish.

## Conclusions

This study concludes that the prevalence of undernutrition among adult chronic kidney patients was found to be high. Stage of disease, cause of disease, recent diagnosis of diabetes mellitus and serum albumin value were found to be significant predictors of undernutrition among the CKD patients.

## Supporting information

**S1 File. Analytical procedures done for albumin determination.**
(DOCX)

**S2 File. Questionnaire English version.**
(DOCX)

**S3 File. Amharic version of questionnaire.**
(DOCX)

## Acknowledgments

Our gratitude goes to supervisors, data collectors, study respondents, senior doctors, renal outpatient nurses, staffs and laboratory technicians in Zewditu Memorial and Saint Paul's hospital.

## Author Contributions

**Conceptualization:** Mahder Asefa, Amene Abebe, Daniel Baza.

**Data curation:** Mahder Asefa, Amene Abebe, Behailu Balcha, Daniel Baza.

**Formal analysis:** Mahder Asefa, Amene Abebe, Behailu Balcha, Daniel Baza.

**Investigation:** Mahder Asefa, Amene Abebe, Behailu Balcha, Daniel Baza.

**Methodology:** Mahder Asefa, Amene Abebe, Daniel Baza.

**Project administration:** Behailu Balcha.

**Resources:** Mahder Asefa.

**Software:** Mahder Asefa, Amene Abebe, Behailu Balcha.

**Supervision:** Mahder Asefa, Amene Abebe, Daniel Baza.

**Validation:** Mahder Asefa, Amene Abebe, Behailu Balcha, Daniel Baza.

**Visualization:** Mahder Asefa, Amene Abebe, Daniel Baza.

**Writing – original draft:** Amene Abebe, Behailu Balcha.

**Writing – review & editing:** Mahder Asefa, Amene Abebe, Behailu Balcha, Daniel Baza.

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
