## [Decision Letter · Decision Letter 0]

15 Jan 2021

PONE-D-20-35996

The magnitude of under nutrition and associated factors among adult chronic kidney disease patients in selected hospitals of Addis Ababa, Ethiopia

PLOS ONE

Dear Dr. Baza,

Thank you for submitting your manuscript to PLOS ONE. After careful consideration, we feel that it has merit but does not fully meet PLOS ONE’s publication criteria as it currently stands. Therefore, we invite you to submit a revised version of the manuscript that addresses the points raised during the review process.

We look forward to receiving your revised manuscript.

Kind regards,

Tauqeer Hussain Mallhi, Ph.D

Academic Editor

PLOS ONE

Journal Requirements:

2. Please amend your manuscript to include your abstract after the title page.

4. Please include a caption for figure 2.

Additional Editor Comments:

Dear Authors, your manuscript has been reviewed by relevant experts. They found manuscript interesting but raised several concerns on its presentation, result interpretation, study limitations and interpretations. Referees are unable to fully assess the manuscript due to syntax and grammar errors throughout the manuscript. I will suggest to please consider the proofread service or Native speaker so this draft could be better assessed by the reviewers.

Reviewers' comments:

Reviewer's Responses to Questions

**Comments to the Author**

1. Is the manuscript technically sound, and do the data support the conclusions?

Reviewer #1: Yes

Reviewer #2: Yes

2. Has the statistical analysis been performed appropriately and rigorously? 

Reviewer #1: Yes

Reviewer #2: Yes

3. Have the authors made all data underlying the findings in their manuscript fully available?

Reviewer #1: Yes

Reviewer #2: Yes

4. Is the manuscript presented in an intelligible fashion and written in standard English?

Reviewer #1: No

Reviewer #2: No

5. Review Comments to the Author

Reviewer #1: Asefa et al have reported the prevalance of protein-energy wasting (PEW) in chronic kidney disease (CKD) patients in 2 local tertiary institutions in Ethiopia. They found that cause, duration, and stage of CKD and serum albumin levels are significant predictors of PEW.

Although the findings are not new, this interesting topic deserves to be published. The paper is unnecessary long, language is poor, methods and discussion sections and tables require significant revision.

I consider there is a lot of re-writing by a native speaker. In discussion section there is insufficient circumspection in both interpretations of the findings reported and their relationship to the literature.

I will be honored to review the renovated form of the paper

Reviewer #2: I read carefully the paper entitled " The magnitude of under nutrition and associated factors among adult chronic kidney disease patients in selected hospitals of Addis Ababa, Ethiopia".

This cross-sectional study aimed to assess the magnitude of under nutrition in chronic kidney disease patients in Addis Ababa.

The research question is clear and important, the methodology used is sound but the manuscript needs major revision for structure, language and presentation of results.

I have some comments that could help the authors improve their manuscript.

Major comments:

1- General: The manuscript should be revised for English and authors need to format their references list.

2-The introduction has so many paragraphs, it should be structured using three paragraphs: what we know, what we still are lacking and the hypothesis of the study.

3-Methodology: authors need to explicitly define "under nutrition" in the methods paragraph.

4-Results: in the tables and in text, authors should specify the unit of each variables: age in years, the income's currency. Table 3 was mentioned before table 2 in the text.

5-Authors need to use their references appropriately and specify in their statement the exact characteristics of patients in the cited reference (for instance reference 7 was mentioned after mentioning hemodialysis patients; however, this reference addresses pre-dialysis patients).

6-The limitations of this study should be thoroughly addressed and cited: the reader will not be satisfied by the statement " These study shares the limitations of facility based cross-sectional study".

Minor comments:

1. Abstract:

-Readers would like to see the number of participants in the paragraph "results" of the abstract.

-What do authors mean by "diabetic patients with glomerulonephritis"?

-In the conclusion: "The prevalence of under nutrition in this study was higher"....higher than?

2. Introduction

-Authors should consider using other terms than "frightening", and "worst stage" and "victims".

6. PLOS authors have the option to publish the peer review history of their article (what does this mean?). If published, this will include your full peer review and any attached files.

Reviewer #1: No

Reviewer #2: **Yes: **Mabel Aoun

---

## [Author Response · Author response to Decision Letter 0]

4 Mar 2021

Response to reviewers

General note for academic editor regarding the overall progress

 First of all, we would like to acknowledge the academic editor for giving us adequate time to revise and address all the concerns of the reviewers and journal requirements. Following, we the authors of this manuscript have been working extensively since we have been notified with the academic editor and expert reviewer’s reports of the manuscript giving a due attention for all the concerns raised by the academic editor and expert reviewers to be well addressed. 

The presentations of the results of the study have been critically worked and made to be in a logical and coherent sequence, study limitations and interpretations are re-written and sufficiently and clearly elaborated. The whole manuscript file has almost been re-written correcting all the syntax and grammatical errors. We hope the academic editor and the expert reviewers get it as much improved manuscript. Thank you so much! 

A. Point by point response letter to academic editor 

1. We have checked again our manuscript for fulfillment of PLOS ONEs style requirements including the naming of files and it has been written accordingly. Thank you!!

2. We have amended the abstract and placed the abstract after the title page as per the recommendation. Thank you!!

3. We placed the ethics approval statement at the end of methods section only and removed it from the prior section of the manuscript. Thank you in advance!!!

4. As per your recommendation, we inserted caption for figure 2 .Thank you in advance!!!

B. Point by point response letter to reviewer one

First of all we would like to express our gratefulness for the reviewer for appropriately recognizing the topic as one of the important area of research and for being interested on the topic. Following our acknowledgement, the reviewers concerns are point by point addressed in the following bullets

The reviewer’s comment regarding the papers being unnecessarily long is well accepted. We make the current revised version clear and focused only to the study objectives. All the other out of the scope of the study which made the paper unnecessarily long have been removed and the study objectives related contents are further and adequately elaborated. We hope the reviewer will get the revised version clear and to the point of the study objectives. Thank you so much!

The reviewer’s comment regarding the language errors are also the right concerns and we have re-written each and every statement throughout the manuscript correcting all the grammatical and punctuation errors consulting English language experts in Wolaita Sodo University. We hope the reviewer will get the revised version is clear and improved. Thank you!

The reviewer’s comments regarding methods, discussion and tables of the previous manuscript require significant revision is well accepted. In the current revised version, we made all the methods followed during the study are well and clearly explained, in the discussion section the important finding are clearly discussed, tables and figures are well revised. Thank you!

The reviewer’s comments about the discussion sections lack of sufficient circumspection both in the interpretation and relationship with the extant literature are right. In the recent revised version we have appropriately re-written the whole discussion section and we made the interpretations clear and discussed the findings with relevant and up to date results in the extant literature. Thank you!

As overall, the recent revised manuscript file is adequately renovated version and we hope the reviewer will get the manuscript as much improved. Thank you so much in advance! 

C. Point by point response letter to reviewer two

Before all, we would also like to express our thankfulness of reviewer two for carefully and critically reviewing this manuscript sacrificing crucial time and effort for maintaining scientific integrity of the manuscript and to be considered for publication meeting the scientific and journal requirements. Following our gratefulness of the reviewer, we have addressed all the reviewers concerns point by point as follows:

Point by point response to major comments of reviewer two

1. The reviewers concerns about revision of the language and formatting reference list is well accepted. Accordingly, we have thoroughly read the whole manuscript file’s each and every statement for any grammatical errors and corrected throughout the document and we have formatted the reference list as it appears in the manuscript document sequentially. Thank you!

2. The reviewers comments regarding the structure of the introduction section of the previous version has been accepted and we re-structured the introduction section of this revised manuscript as per the reviewers recommendation in to three parts “what have been known in the existing body of knowledge”, “what is lacking in the current literature” and “the hypothesis of the this particular study”. Thank you so much! 

3. The reviewer’s comment to explicitly define “undernutrition” in methods section is accepted and as per the reviewers recommendation we have explicitly defined undernutrition and other relevant concepts in the methods section of the manuscript. Thank you! 

4. The reviewer’s comments regarding the some important variables lack of unit of measurement (age in years and currency) and mentioning Table 3 before Table 2 have been accepted. We have included the units of measurements for age and currency and all the tables are sequentially placed in the manuscript file. Thank you!

5. The authors would like to thank the reviewer for the best insight regarding incomparable referencing of the characteristics of CKD patients. Reference number 6 has been removed from the text and reference list, because the study was on CKD patients who were on hemodialysis which is not in the scope of this study and the concept held in reference number 7 has been elaborated further because it is a matched reference with the current study. Thank you!

6. The reviewer’s comments about unsatisfactorily addressing of the limitations of the study in the previous version are right and well accepted. In the current revised version, we have appropriately and satisfactorily addressed the limitations of the study including all the possible limitations of this particular study. We hope the reviewer will get the concerns are well addressed. Thank you in advance!

Point by point response to minor comments of reviewer two

1. The reviewer’s comment regarding the number of participants to be included in the abstract results part is right and it has been included. Thank you!

2. According to the reviewers comment, we noticed the phrase “diabetic glomerulonephritis” was unnecessary and irrelevant to this particular study and it has been removed from the manuscript file. Instead, we have operationalized the term “glomerulonephritis” which is an important variable for the study and included in operational definition section of the main manuscript file. Thank you!

3. The reviewers comments regarding the comparison of the “prevalence of undernutrition among chronic kidney disease patients” in the abstract conclusion section is accepted. As per the suggestion of the reviewer in the current revised version the “prevalence of undernutrition among chronic kidney disease patients” has been described relative to its public health significance when compared to its prevalence in the Sub-Saharan Africa region. Thank you!

4. The reviewer’s comment regarding the words which are unnecessarily terrifying has been accepted and replaced with appropriate ones. Thank you!

Thank you!

---

## [Decision Letter · Decision Letter 1]

6 Apr 2021

PONE-D-20-35996R1

The magnitude of undernutrition and associated factors among adult chronic kidney disease patients in selected hospitals of Addis Ababa, Ethiopia

PLOS ONE

Dear Dr. Baza,

Thank you for submitting your manuscript to PLOS ONE. After careful consideration, we feel that it has merit but does not fully meet PLOS ONE’s publication criteria as it currently stands. Therefore, we invite you to submit a revised version of the manuscript that addresses the points raised during the review process.

We look forward to receiving your revised manuscript.

Kind regards,

Tauqeer Hussain Mallhi, Ph.D

Academic Editor

PLOS ONE

Additional Editor Comments (if provided):

Thank for submitting the revised version. However, manuscript requires further improvement as suggested by the reviewer such as methodology, English/grammatical corrections, and statistical analysis. Please incorporate the comments of the reviewer so we could reach an appropriate decision.

Reviewers' comments:

Reviewer's Responses to Questions

**Comments to the Author**

1. If the authors have adequately addressed your comments raised in a previous round of review and you feel that this manuscript is now acceptable for publication, you may indicate that here to bypass the “Comments to the Author” section, enter your conflict of interest statement in the “Confidential to Editor” section, and submit your "Accept" recommendation.

Reviewer #1: All comments have been addressed

Reviewer #2: (No Response)

2. Is the manuscript technically sound, and do the data support the conclusions?

Reviewer #1: Yes

Reviewer #2: Partly

3. Has the statistical analysis been performed appropriately and rigorously? 

Reviewer #1: Yes

Reviewer #2: I Don't Know

4. Have the authors made all data underlying the findings in their manuscript fully available?

Reviewer #1: Yes

Reviewer #2: No

5. Is the manuscript presented in an intelligible fashion and written in standard English?

Reviewer #1: Yes

Reviewer #2: No

6. Review Comments to the Author

Reviewer #1: As I mentioned in my first review, although the findings are not new, this topic deserves to be published. After major revisions, the paper looks well. I consider that it can be accepted in its current form.

Reviewer #2: Dear editor,

Thank you for the opportunity to read the revised version of the manuscript " The magnitude of undernutrition and associated factors among adult chronic kidney disease patients in selected hospitals of Addis Ababa, Ethiopia".

The authors put some efforts in order to improve their paper, however it still needs further work.

1-Vocabulary and grammar.

-In the abstract: comorbid is an adjective and not a noun; you can say "comorbid condition" or "comorbidity". "This study aimed " and not "was aimed".

-In the introduction: evidence not evidences. "Fasten" does not mean makes it faster in English! What do you mean by "primary complication"? An important complication?

Authors used ESRD and ESKD: one nomenclature is less confusing to the reader.

-In the methodology: patients were actually "included" not "participated".

-In the discussion: "has described" not "had". 160 has no meaning without the total: 160 out of...and it is not a magnitude, it is a prevalence.

Third sentence of the discussion lacks a verb.

Etc, etc.

2-Minor comments:

-In the abstract, results' section: Better to remove 160 and leave 43.1% for the prevalence.

-The definition of glomerulonephritis is usually not biological.

3-Major comments:

-Why did the author define hypoalbuminemia as less than 3.8 and not 3.5 g/dL? If you take the threshold of 3.5 as hypoalbuminemia, would hypoalbuminemia be associated with the low BMI?

-The interpretation of the results of the regression analysis should be different. The conclusion should state that a factor is associated to the outcome (dependent variable) and it is not a comparison of groups.

-In the whole manuscript including abstract, it is not explicitly said in the methodology that undernutrition is defined as a low BMI. What is your definition of undernutrition? BMI<18.5? This should be clearly stated in the methods.

-In the strengths of the study: what do authors mean by "considering the serum albumin to see the anemia clinically".

7. PLOS authors have the option to publish the peer review history of their article (what does this mean?). If published, this will include your full peer review and any attached files.

Reviewer #1: No

Reviewer #2: **Yes: **Mabel Aoun

---

## [Author Response · Author response to Decision Letter 1]

23 Apr 2021

General note to academic editor

First of all, we would like to express gratitude for the academic editor for giving us sufficient time to address all concerns of the reviewer. Following, our acknowledgment we have been working extensively since we have been notified to address the concerns of the expert reviewer and explained point by point in the following bullets. 

Point by point response for reviewer 2

• Before all, we would like to express our thankfulness of reviewer two for carefully and critically reviewing revised manuscript. We found the comments and recommendations of the reviewer are very important inputs to maintain standards of the journal requirements for publication. 

1. Point by point response to vocabulary and grammar errors for reviewer 

• The reviewer comment regarding vocabulary and grammar errors are the right concerns. We have incorporated the recommendations of the reviewer and also checked the whole manuscript for similar errors and corrected accordingly. We hope the reviewer can access all the revisions in the clean copy of the manuscript and also from the revised manuscript with track changes. Thank you in so much!

2. Point by point response to minor comments of reviewer 2

• The reviewer comment regarding to remove 160 and leave 43.1% for the prevalence is acceptable and we have corrected accordingly. Thank you!

• The concern of the reviewer about the definition of glomerulonephritis is right. However, we defined it as it appears in the studied hospitals diagnostic criteria. Thank you!

3. Point by point response to major comments of reviewer 2

• The reviewer concern regarding the cut-point off hypoalbuminemia is appropriate and right. However, when we collect the information for the study, we have raised the concern with the clinicians on follow up care for the CKD patients. The normal serum albumin for general population ranges from 3.5g/dl- 5.2g/dl. The minimum cut- off value of 3.8g/dl was being used as a lower limit to consider hypoalbuminemia in the studied hospitals because CKD patients are vulnerable group for nutritional anemia. To avoid any delays in nutritional interventions for the CKD patients, the lower cut-off limit has been raised from 3.5g/dl to 3.8g/dl. This is why the studied hospitals are using 3.8g/dl as lower limit to consider hypoalbuminemia and we have used the same lower limit for the study. Thank you so much!

• According to the reviewer suggestion we have removed the comparison of the findings from the conclusion section and associated factors with the dependent variable has been stated in the current revised version of the manuscript. Thank you!

• The reviewer comment regarding explicit definition of the undernutrition in the abstract and the method sections is appropriate and acceptable. We have explicitly defined undernutrition as BMI<18.5 in both sections. Thank you!

• The reviewer comment regarding the strength of the study section is right. We have revised the section in the recent revised version and made clear. Thank you!

---

## [Decision Letter · Decision Letter 2]

3 May 2021

The magnitude of undernutrition and associated factors among adult chronic kidney disease patients in selected hospitals of Addis Ababa, Ethiopia

PONE-D-20-35996R2

Dear Dr. Baza,

We’re pleased to inform you that your manuscript has been judged scientifically suitable for publication and will be formally accepted for publication once it meets all outstanding technical requirements.

Kind regards,

Tauqeer Hussain Mallhi, Ph.D

Academic Editor

PLOS ONE

Additional Editor Comments (optional):

Reviewers' comments:

Reviewer's Responses to Questions

**Comments to the Author**

1. If the authors have adequately addressed your comments raised in a previous round of review and you feel that this manuscript is now acceptable for publication, you may indicate that here to bypass the “Comments to the Author” section, enter your conflict of interest statement in the “Confidential to Editor” section, and submit your "Accept" recommendation.

Reviewer #1: All comments have been addressed

Reviewer #2: (No Response)

2. Is the manuscript technically sound, and do the data support the conclusions?

Reviewer #1: Yes

Reviewer #2: Yes

3. Has the statistical analysis been performed appropriately and rigorously? 

Reviewer #1: Yes

Reviewer #2: Yes

4. Have the authors made all data underlying the findings in their manuscript fully available?

Reviewer #1: Yes

Reviewer #2: Yes

5. Is the manuscript presented in an intelligible fashion and written in standard English?

Reviewer #1: No

Reviewer #2: No

6. Review Comments to the Author

Reviewer #1: The main subject of the manuscript deserves to be published. After language revision, the paper looks better. It can be accepted.

Reviewer #2: (No Response)

7. PLOS authors have the option to publish the peer review history of their article (what does this mean?). If published, this will include your full peer review and any attached files.

Reviewer #1: No

Reviewer #2: No

---

## [Editor Report · Acceptance letter]

29 Jun 2021

PONE-D-20-35996R2 

The magnitude of undernutrition and associated factors among adult chronic kidney disease patients in selected hospitals of Addis Ababa, Ethiopia 

Dear Dr. Baza:

I'm pleased to inform you that your manuscript has been deemed suitable for publication in PLOS ONE. Congratulations! Your manuscript is now with our production department. 

Kind regards, 

on behalf of

Dr. Tauqeer Hussain Mallhi 

Academic Editor

PLOS ONE